# MULTISCALE SCORE MATCHING FOR OUT-OF-DISTRIBUTION DETECTION

**Ahsan Mahmood, Junier Oliva, Martin Styner**
Department of Computer Science
University of North Carolina at Chapel Hill
{amahmood, joliva, styner}@cs.unc.edu

## ABSTRACT

We present a new methodology for detecting out-of-distribution (OOD) images by utilizing norms of the score estimates at multiple noise scales. A score is defined to be the gradient of the log density with respect to the input data. Our methodology is completely unsupervised and follows a straight forward training scheme. First, we train a deep network to estimate scores for $L$ levels of noise. Once trained, we calculate the noisy score estimates for $N$ in-distribution samples and take the L2-norms across the input dimensions (resulting in an $N$x$L$ matrix). Then we train an auxiliary model (such as a Gaussian Mixture Model) to learn the in-distribution spatial regions in this $L$-dimensional space. This auxiliary model can now be used to identify points that reside outside the learned space. Despite its simplicity, our experiments show that this methodology significantly outperforms the state-of-the-art in detecting out-of-distribution images. For example, our method can effectively separate CIFAR-10 (inlier) and SVHN (OOD) images, a setting which has been previously shown to be difficult for deep likelihood models. We make our code and results publicly available on Github [1].

## 1 INTRODUCTION

Modern neural networks do not tend to generalize well to out-of-distribution samples. This phenomenon has been observed in both classifier networks (Hendrycks & Gimpel (2017); Nguyen et al. (2015); Szegedy et al. (2013)) and deep likelihood models (Nalisnick et al. (2018); Hendrycks et al. (2018); Ren et al. (2019)). This certainly has implications for AI safety (Amodei et al. (2016)), as models need to be aware of uncertainty when presented with unseen examples. Moreover, an out-of-distribution detector can be applied as an anomaly detector. Ultimately, our research is motivated by the need for a sensitive outlier detector that can be used in a medical setting. Particularly, we want to identify atypical morphometry in early brain development. This requires a method that is generalizable to highly variable, high resolution, unlabeled real-world data while being sensitive enough to detect an unspecified, heterogeneous set of atypicalities. To that end, we propose *multiscale score matching* to effectively detect out-of-distribution samples.

Hyvärinen (2005) introduced score matching as a method to learn the parameters of a non-normalized probability density model, where a score is defined as the gradient of the log density with respect to the data. Conceptually, a score is a vector field that points in the direction where the log density grows the most. The authors mention the possibility of matching scores via a non-parametric model but circumvent this by using gradients of the score estimate itself. However, Vincent (2011) later showed that the objective function of a denoising autoencoder (DAE) is equivalent to matching the score of a non-parametric Parzen density estimator of the data. Thus, DAEs provide a methodology for learning score estimates via the objective:

$$\frac{1}{2} \mathbb{E}_{\tilde{x} \sim q_\sigma(\tilde{x}|x)p_{\text{data}}(x)}[||s_\theta(\tilde{x}) - \nabla_{\tilde{x}} \log q_\sigma(\tilde{x}|x)||] \tag{1}$$

Here $s_\theta(x)$ is the score network being trained to estimate the true score $\nabla_x \log p_{\text{data}}(x)$, and $q_\sigma(\tilde{x}) = \int q_\sigma(\tilde{x}|x)p_{\text{data}}(x)dx$. It should be noted that the score of the estimator only matches

---

[1] https://github.com/ahsanMah/msma

the true score when the noise perturbation is minimal i.e $q_\sigma(\tilde{x}) \approx p_{\text{data}}(x)$. Recently, Song & Ermon (2019) employed multiple noise levels to develop a deep generative model based on score matching, called Noise Conditioned Score Network (NCSN). Let $\{\sigma_i\}_{i=1}^L$ be a positive geometric sequence that satisfies $\frac{\sigma_1}{\sigma_2} = ... = \frac{\sigma_{L-1}}{\sigma_L} > 1$. NCSN is a conditional network, $s_\theta(x, \sigma)$, trained to jointly estimate scores for various levels of noise $\sigma_i$ such that $\forall \sigma \in \{\sigma_i\}_{i=1}^L : s_\theta(x, \sigma) \approx \nabla_x \log q_\sigma(x)$. In practice, the network is explicitly provided a one-hot vector denoting the noise level used to perturb the data. The network is then trained via a denoising score matching loss. They choose their noise distribution to be $\mathcal{N}(\tilde{x}|x, \sigma^2 I)$; therefore $\nabla_{\tilde{x}} \log q_\sigma(\tilde{x}|x) = -(\tilde{x} - x/\sigma^2)$. Thus the objective function is:

$$\frac{1}{L} \sum_{i=1}^L \lambda(\sigma_i) \left[ \frac{1}{2} \mathbb{E}_{\tilde{x} \sim q_{\sigma_i}(\tilde{x}|x) p_{\text{data}}(x)} \left[ \left|\left| s_\theta(\tilde{x}, \sigma_i) + (\frac{\tilde{x} - x}{\sigma_i^2}) \right|\right|_2^2 \right] \right] \tag{2}$$

Song & Ermon (2019) set $\lambda(\sigma_i) = \sigma^2$ after empirically observing that $||\sigma s_\theta(x, \sigma)||_2 \propto 1$. We similarly scaled our score norms for all our experiments. Our work directly utilizes the training objective proposed by Song & Ermon (2019) i.e. we use an NCSN as our score estimator. However, we use the score outputs for out-of-distribution (OOD) detection rather than for generative modeling. We demonstrate how the space of multiscale score estimates can separate in-distribution samples from outliers, outperforming state-of-the-art methods. We also apply our method on real-world medical imaging data of brain MRI scans.

## 2 MULTISCALE SCORE ANALYSIS

Consider taking the L2-norm of the score function: $||s(x)|| = ||\nabla_x \log p(x)|| = \left|\left| \frac{\nabla_x p(x)}{p(x)} \right|\right|$.

Since the data density term appears in the denominator, a high likelihood will correspond to a low norm. Since out-of-distribution samples should have a low likelihood with respect to the in-distribution log density (i.e. $p(x)$ is small), we can expect them to have high score norms. However, if these outlier points reside in "flat" regions with very small gradients (e.g. in a small local mode), then their score norms can be low despite the point belonging to a low density region. This is our first indicator informing us that a true score norm may not be sufficient for detecting outliers. We empirically validate our intuition by considering score estimates for a relatively simple toy dataset: FashionMNIST. Following the denoising score matching objective (Equation 2), we can obtain multiple estimates of the true score by using different noise distributions $q_\sigma(\tilde{x}|x)$. Like Song & Ermon (2019), we choose the noise distributions to be zero-centered Gaussian scaled according to $\sigma_i$. Recall that the scores for samples perturbed by the lowest $\sigma$ noise should be closest to the true score. Our analyses show that this alone was inadequate at separating inliers from OOD samples.

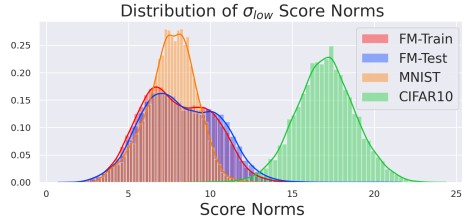
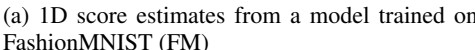
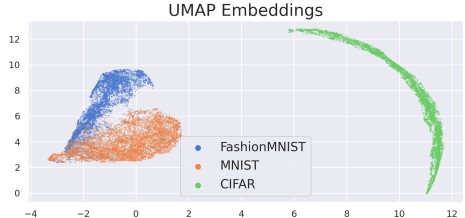

(a) 1D score estimates from a model trained on FashionMNIST (FM)

(b) UMAP visualization of 10-dimensional score estimates from a model trained on FashionMNIST

Figure 1: Visualizing the need for a multiscale analysis. In (a), we plot the scores corresponding to the lowest sigma estimate. In (b), we plot the UMAP embedding of the $L = 10$ dimensional vectors of score norms. Here we see a better separation between FashionMNIST and MNIST when using estimates from *multiple* scales rather than the one that corresponds to the true score only.

We trained a score network $s_{\text{FM}}(x, \sigma_L)$ on FashionMNIST and used it to estimate scores of FashionMNIST ($x \sim D_{FM}$), MNIST ($x \sim D_M$) and CIFAR-10 ($x \sim D_C$) test sets. Figure 1a shows the distribution of the score norms corresponding to the lowest noise level used. Note that CIFAR-10 samples are appropriately given a high score by the model. However, the model is unable to

distinguish FashionMNIST from MNIST, giving MNIST roughly the same scores as in-distribution samples. Though far from ideal, this result is still a considerable improvement on existing likelihood methods, which have been shown to assign *higher* likelihoods to OOD samples (Nalisnick et al. (2018)). Our next line of inquiry was to utilize multiple noise levels. That is instead of simply considering $s_{\text{FM}}(x, \sigma_L)$, we analyze the $L$-dimensional space $[||s_{\text{FM}}(x, \sigma_1)||, ..., ||s_{\text{FM}}(x, \sigma_L)||]$ for $x \sim \{D_{FM}, D_M, D_C\}$. Our observations showed that datasets did tend to be separable in the $L$-dimensional space of score norms. Figure 1b visualizes the UMAP embeddings of scores calculated via a network trained to estimate $L = 10$ scales of $\sigma$s, with the lowest $\sigma$ being the same as one in Figure 1a.

## 2.1 SCALES AND NEIGHBORHOODS

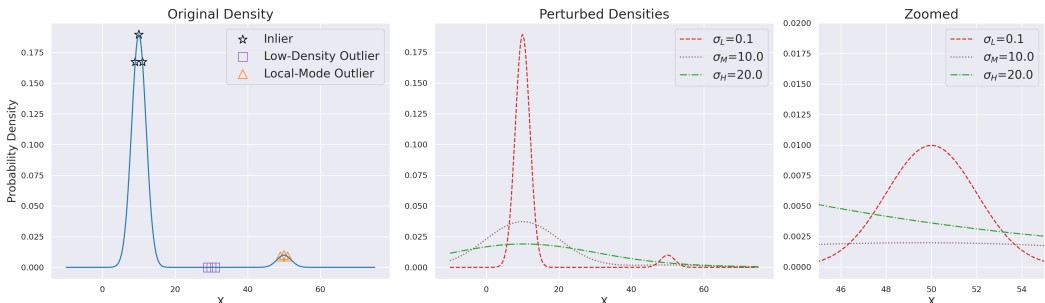

Figure 2: A toy GMM to visualize our analysis. We annotate the three regions of interest we will be exploring. Further, we show the Gaussian perturbed versions of the original distribution with (L)ow, (M)edium, and (H)igh noise levels, along with a plot zoomed into the local-mode outliers. Note the effect of different scales on this region: only the largest scale results in a gradient in the direction of the inliers.

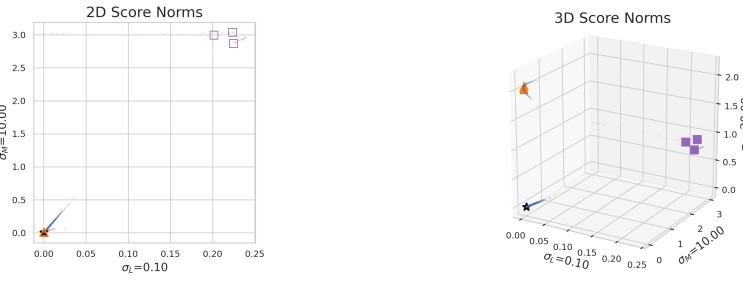

Figure 3: In (a) observe that Low-Density outliers have comparatively high gradient norms for both $\sigma_L$ and $\sigma_M$. However at this scale, Local-Mode points still have very small norms, causing them to be tightly packed around the in-distribution points. In (b) we see that Local-Mode outliers achieve a gradient signal only when a sufficiently high scale is used, $\sigma_H = 20$.

To our knowledge multiscale score analysis has not been explored in the context of OOD detection. In this section, we present an analysis in order to give an intuition for why multiple scales can be beneficial. Consider the toy distribution shown in Figure 2. We have three regions of interest: an inlier region with high density centered around $x = 10$, an outlier region with low density around $x = 30$, and a second outlier region with a local mode centered around $x = 50$. Recall that adding Gaussian noise to a distribution is equivalent to convolving it with the Gaussian distribution. This not only allows us to visualize perturbations of our toy distribution, but also analytically compute the score estimates given any $\sigma_i$. Initially with no perturbation, both a point in the low-density region and one very close to (or at) the local-mode will have small gradients. As we perturb the samples we smooth the original density, causing it to widen. The relative change in density at each point is dependent on neighboring modes. A large scale perturbation will proportionally take a larger neighborhood into account at each point of the convolution. Therefore, at a sufficiently large scale,

nearby outlier points gain context from in-distribution modes. This results in an increased gradient signal in the direction of inliers.

Figure 3 plots the score norms of samples generated from the original density along with markers indicating our key regions. Note how even a small scale perturbation ($\sigma_L = 0.1$) is enough to bias the density of the Low-Density outliers towards the nearby in-distribution mode. A medium scale ($\sigma_M = 10$) Gaussian perturbation is still not wide enough to reach the inlier region from the Local-Mode outlier densities, causing them to simply smooth away into flat nothingness. It is only after we perform a large scale ($\sigma_H = 20$) perturbation that the in-distribution mode gets taken into account, resulting in a higher gradient norm. This analysis allows us to intuit that larger noise levels account for a larger neighborhood context. We surmise that given a sufficiently large scale, we can capture gradient signals from distant outliers.

It is imperative to note that one large scale is not guaranteed to work for all outliers. Consider outliers close to inlier modes such as the samples between Low-Density outliers and Inliers in Figure 2. $\sigma_H$ results in an overlap in the score distribution of inliers and Low-Density outliers. This makes it difficult to differentiate the aforementioned "in-between" outliers from the in-distribution samples. However, this large scale was necessary to get a big enough neighborhood context in order to capture the more distant Local-Mode outliers. Therefore, we postulate that a *range* of scales is necessary for separating outliers. Admittedly, selecting this range according to the dataset is not a trivial problem. In a very recent work, Song & Ermon (2020) outlined some techniques for selecting $\{\sigma_i\}_{i=1}^L$ for NCSNs from the perspective of generative modelling. Perhaps there is a similar analog to OOD detection. We leave such analyses for future work and use the default range for NCSN in all our experiments. However, we observed that our defaults are surprisingly generalizable, evident by the fact that *all* our experiments in Section 5 were performed with the same scale range. In Section 5.5, we further analyze how varying the scale range effects downstream accuracy and observe that our defaults already provide near optimal performance.

## 2.2 PROPOSED TRAINING SCHEME

In this work, we propound the inclusion of all noisy score estimates for the task of separating in- and out-of-distribution points, allowing for a Multiscale Score Matching Analysis (MSMA). Concretely, given $L$ noise levels, we calculate the L2-norms of per-sample scores for each level, resulting in an $L$-dimensional vector for each input sample. Motivated by our observations, we posit that in-distribution data points occupy distinct and dense regions in the $L$-dimensional score space. The *cluster assumption* states that decision boundaries should not pass high density regions, but instead lie in low density regions. This implies that any auxiliary method trained to learn in-distribution regions should be able to identify OOD data points that reside outside the learned space. Thus, we propose a two step unsupervised training scheme. First, we train a NCSN model $s_{\text{IN}}(x, \sigma_L)$ to estimate scores for inlier samples, given $\{\sigma_i\}_{i=1}^L$ levels of noise. Once trained, we calculate all $L$ noisy score estimates for the $N$ training samples and take the L2-norms across the input dimensions: $[||s_{\text{IN}}(X, \sigma_1)||_2^2, ..., ||s_{\text{IN}}(X, \sigma_L)||_2^2]$. This results in an $N$x$L$ matrix. We now train an auxiliary model (such as a Gaussian Mixture Model) on this matrix to learn the spatial regions of in-distribution samples in the $L$-dimensional space.

## 3 LEARNING CONCENTRATION IN THE SCORE SPACE

We posit that learning the "density" of the inlier data in the $L$-dimensional score (norm) space is sufficient for detecting out-of-distribution samples. The term "density" can be interpreted in a myriad of ways. We primarily focus on models that fall under three related but distinct notions of denseness: spatial clustering, probability density, and nearest (inlier) neighbor graphs. All three allow us to threshold the associated metric to best separate OOD samples.

Spatial clustering is conceptually the closest to our canonical understanding of denseness: points are tightly packed under some metric (usually Euclidean distance). Ideally OOD data should not occupy the same cluster as the inliers. We train Gaussian Mixture Models (GMMs) to learn clusters in the inlier data. GMMs work under the assumption that the data is composed of k-components whose shapes can be described by a (multivariate) Gaussian distribution. Thus for a given datum, we can calculate the joint probability of it belonging to any of the k-components.

Probability density estimation techniques aim to learn the underlying probability density function $p_{data}(x)$ which describes the population. Normalizing flows are a family of flexible methods that can learn tractable density functions (Papamakarios et al. (2019)). They transform complex distributions into a simpler one (such as a Gaussian) through a series of invertible, differential mappings. The simpler base distribution can then be used to infer the density of a given sample. We use Masked Autoregressive Flows introduced by Papamakarios et al. (2017), which allows us to use neural networks as the transformation functions. Once trained, we can use the likelihood of the inliers to determine a threshold after which samples will be considered outliers.

Finally, we consider building k-nearest neighbor (k-NN) graphs to allow for yet another thresholding metric. Conceptually, the idea is to sort all samples according to distances to their k-closest (inlier) neighbor. Presumably, samples from the same distribution as the inliers will have very short distances to training data points. Despite its simplicity, this method works surprisingly well. Practically, k-NN distances can be computed quickly compute by using efficient data structures (such as KD Trees).

## 4 RELATED WORK

Hendrycks & Gimpel (2017) should be commended for creating an OOD baseline and establishing an experimental test-bed which has served as a template for all OOD work since. Their purported method was thresholding of softmax probabilities of a well-trained classifier. Their results have since been beaten by more recent work. Liang et al. (2017) propose ODIN as a post-hoc method that utilizes a pretrained network to reliably separate OOD samples from the inlier distribution. They achieve this via i) perturbing the input image in the gradient direction of the highest (inlier) softmax probability and ii) scaling the temperature of the softmax outputs of the network for the best OOD separation. They follow the setting from Hendrycks & Gimpel (2017) and show very promising results for the time. However, ODIN heavily depends on careful tuning of its hyperparameters

DeVries & Taylor (2018) train their networks to predict confidence estimates in addition to softmax probabilities, which can then be used to threshold outliers. They show significant improvements over Hendrycks & Gimpel (2017) and some improvements over ODIN. Another concurrent work by Lee et al. (2018) jointly trained a GAN alongside the classifier network to generate 'realistic' OOD examples, requiring an additional OOD set during training time. The final trained network is also unable to generalize to other unseen datasets. It is important to note that our method is trained completely unsupervised while the baselines are not, potentially giving them additional information about the idiosyncrasies of the inlier distribution.

Ren et al. (2019) proposed to jointly train deep likelihood models alongside a "background" likelihood model that learns the population level background statistics, taking the ratio of the two resulting likelihoods to produce a "contrastive score". They saw very good results for grayscale images (FashionMnist vs MNIST) and saw a considerable improvement in separating CIFAR and SVHN compared to Nalisnick et al. (2018). Some prior work has indeed used gradients of the log likelihoods for OOD but they do not frame it in the context of score matching. Grathwohl et al. (2020) posits that a discriminative model can be reinterpreted as a joint energy (negative loglikelihood) based model (JEM). One of their evaluation experiments used the energy norms (which they dub 'Approximate Mass JEM') for OOD detection. Even though they saw improvements over only using log-likelihoods, their reported AUCs did not beat ODIN or other competitors. Peculiarly, they also observed that for tractable likelihood models, scores were anti-correlated with the model's likelihood and that neither were reliable for OOD detection. Zhai et al. (2016) also used energy (negative log probability) gradient norms, but their experiments were limited to intra-dataset anomalies. To our knowledge, no prior work has explicitly used score matching for OOD detection.

## 5 EXPERIMENTS

In this section we demonstrate MSMA's potential as an effective OOD detector. We first train a NCSN model as our score estimator, and then an auxiliary model on the score estimates of the training set. Following Liang et al. (2017) and DeVries & Taylor (2018), we use CIFAR-10 and SVHN as our "inlier" datasets alongside a collection of natural images as "outlier" datasets. We

retrieve the natural image from ODIN's publicly available GitHub repo[2]. This helps maintain a fair comparison (e.g. it ensures we test on the same random crops as ODIN). We will denote Liang et al. (2017) as ODIN and DeVries & Taylor (2018) as Confidence in our tables. We also distinguish *between* CIFAR and SVHN and compare our results to the state-of-the-art. Additionally, we report our results for FashionMNIST vs MNIST in Section A.6.

## 5.1 DATASETS AND EVALUATION METRICS

We consider CIFAR-10 (Krizhevsky et al. (2009)) and SVHN (Netzer et al. (2011)) as our inlier datasets. For out-of-distribution datasets, we choose the same as Liang et al. (2017): **TinyImageNet**[3], **LSUN** (Yu et al. (2015)), **iSUN** (Xu et al. (2015)). Similar to DeVries & Taylor (2018), in our main experiments we report only *resized* versions of *LSUN* and *TinyImageNet*. We also leave out the synthetic **Uniform** and **Gaussian** noise samples from our main experiments as we performed extremely well in all of those experiments. We refer the reader to A.4 for the full report including all datasets. Lastly following DeVries & Taylor (2018), we consider **All Images**: a combination of all non-synthetic OOD datasets outlined above (including *cropped* versions). Note that this collection effectively requires a single threshold for all datasets, thus arguably reflecting a real world out-of-distribution setting. To measure thresholding performance we use the metrics established by previous baselines (Hendrycks & Gimpel (2017), Liang et al. (2017)). **FPR at 95% TPR** is the False Positive Rate (FPR) when the True Positive Rate (TPR) is 95%. **Detection Error** is the minimum possible misclassification probability over all thresholds. **AUROC** is Area Under the ROC Curve. **AUPR** is Area Under the Precision Recall Curve. More details are given in A.3.

## 5.2 COMPARISON AGAINST PREVIOUS OOD METHODS

We compare our work against Confidence Thresholding (DeVries & Taylor (2018)) and ODIN (Liang et al. (2017)). For all experiments we report the results for the in-distribution *testset* vs the out-of-distribution datasets. Note that *All Images\** is a version of *All Images* where both ODIN and Confidence Thresholding perform input preprocessing. Particularly, they perturb the samples in the direction of the softmax gradient of the classifier: $\tilde{x} = x - \epsilon \, \text{sign}(-\nabla_x log S_y(x; T))$. They then perform a grid search over $\epsilon$ ranges, selecting the value that achieves best separation on 1000 samples randomly held for *each* out-of-distribution set. ODIN performs an additional search over $T$ ranges, while Confidence Thresholding uses a default of $T = 1000$. We do not perform any such input modification. Note that ODIN uses input thresholding for individual OOD datsets as well, while Confidence Thresholding does not. Finally, for the sake of brevity we only report **FPR (95% TPR)** and **AUROC**. All other metric comparisons are available in the appendix (A.4).

| Dataset | OOD Dataset | GMM MSMA | Flow MSMA | KD Tree MSMA | ODIN | Confidence |
|---------|-------------|----------|-----------|--------------|------|------------|
| SVHN | TinyImageNet | **0.0 / 100.0** | **0.0 / 100.0** | **0.0 / 100.0** | - | 1.8 / 99.6 |
|  | LSUN | **0.0 / 100.0** | **0.0 / 100.0** | **0.0 / 100.0** | - | 0.8 / 99.8 |
|  | iSUN | **0.0 / 100.0** | **0.0 / 100.0** | **0.0 / 100.0** | - | 1.0 / 99.8 |
|  | All Images | **0.0 / 100.0** | **0.0 / 100.0** | **0.0 / 100.0** | - | 4.3 / 99.2 |
|  | All Images* | - | - | - | 8.6 / 97.2 | 4.1 / 99.2 |
| CIFAR-10 | TinyImageNet | **0.0 / 100.0** | **0.0 / 100.0** | 0.3 / 99.9 | 7.5 / 98.5 | 18.4 / 97.0 |
|  | LSUN | **0.0 / 100.0** | **0.0 / 100.0** | 0.6 / 99.9 | 3.8 / 99.2 | 16.4 / 97.5 |
|  | iSUN | **0.0 / 100.0** | **0.0 / 100.0** | 0.4 / 99.9 | 6.3 / 98.8 | 16.3 / 97.5 |
|  | All Images | **0.0 / 100.0** | **0.0 / 100.0** | 0.4 / 99.9 | - | 19.2 / 97.1 |
|  | All Images* | - | - | - | 7.8 / 98.4 | 11.2 / 98.0 |

Table 1: Comparing our auxiliary methods against existing state-of-the-art. **FPR (95% TPR)** (*Lower* is better)/ **AUROC** (*Higher* is better). MSMA methods unequivocally outperform previous work in all tests, with KD Trees only slightly worse than GMM and Flow variants.

---

[2] https://github.com/facebookresearch/odin
[3] https://tiny-imagenet.herokuapp.com/

## 5.3 SEPARATING CIFAR-10 FROM SVHN

Since this setting (CIFAR-10 as in-distribution and SVHN as out-of-distribution) is not considered in the testbed used by ODIN or Confidence Thresholding, we consider these results separately and evaluate them in the context of likelihood methods. This experiment has recently gained attention following Nalisnick et al. (2018) showing how deep generative models are particularly inept at separating high dimensional complex datasets such as these two. We describe our results for each auxiliary model in Table 2. Here we note that *all* three methods definitively outperform the previous state of the art (see Table 3), with KD-Trees preforming the best. Likelihood Ratios Ren et al. (2019) JEM (Grathwohl et al. (2020)) are two unsupervised methods that have tackled this problem and have reported current state-of-the-art results. Table 3 summarizes the results that were reported by these papers. Both report AUROCs, with Ren et al. (2019) additionally reporting AUPR(In) and FPR at 80% TPR. Since each method proposes a different detection function, we also provide them for reference.

| Auxiliary Method | FPR (95% TPR) ↓ | Detection Error ↓ | AUROC ↑ | AUPR In ↑ | AUPR Out ↑ |
|---|---|---|---|---|---|
| GMM | 11.4 | 8.1 | 95.5 | 91.9 | 96.9 |
| Flow | 8.6 | 6.8 | 96.7 | 93.4 | 97.7 |
| KD Tree | **4.1** | **4.5** | **99.1** | **99.1** | **99.2** |

Table 2: Comparison of auxiliary models tasked to separate CIFAR-10 (inlier) and SVHN (out-of-distribution). ↓ indicates lower values are better and ↑ indicates higher values are better.

| Detection Function | Model | FPR (80% TPR) ↓ | AUROC ↑ | AUPR In ↑ | AUPR Out ↑ |
|---|---|---|---|---|---|
| $\|s_\theta(x, \sigma_i^L)\|$ | KD Tree MSMA | **0.7** | **99.1** | **99.1** | **99.2** |
| $\log \frac{p_\theta(x)}{p_{\theta_0}(x)}$ | Likelihood Ratios | 6.6 | 93.0 | 88.1 | - |
| $\log p(x)$ | JEM | - | 67.0 | - | - |
| $\left\|\left\|\frac{\partial \log p(x)}{\partial x}\right\|\right\|$ | Approx. Mass JEM | - | 83.0 | - | - |

Table 3: Comparison with a multitude of likelihood-based models at separating CIFAR-10 (in-distribution) from SVHN (out-of-distribution). - represent metrics that were not reported by the work. All values are shown in percentages. ↓ indicates lower values are better and ↑ indicates higher values are better. Note that since Likelihood Ratios report FPR at **80%** TPR, we report the same.

## 5.4 AGE BASED OOD FROM BRAIN MRI SCANS

| OOD Age (Years) | FPR (95% TPR) ↓ | Detection Error ↓ | AUROC ↑ | AUPR In ↑ | AUPR Out ↑ |
|---|---|---|---|---|---|
| | | | Baseline( Hendrycks & Gimpel (2017)) / MSMA | | |
| 1 | **0.0** / 0.2 | **0.0** / 0.4 | **100.0** / 99.9 | **100.0** / 99.9 | **100.0** / 99.9 |
| 2 | 41.9 / **0.6** | 20.1 / **1.0** | 64.8 / **99.7** | 70.3 / **99.5** | 77.6 / **99.9** |
| 4 | 45.0 / **23.7** | 15.8 / **9.2** | 80.0 / **96.1** | 81.1 / **93.8** | 84.8 / **97.9** |
| 6 | 44.9 / **30.5** | 18.7 / **9.7** | 81.5 / **95.0** | 80.7 / **92.2** | 86.6 / **96.8** |

Table 4: MSMA-GMM trained on multiscale score estimates tasked to separate the brain scans of different age groups. In-distribution samples are 9-11 years of age. All values are shown in percentages. ↓ indicates lower values are better and ↑ indicates higher values are better. Keep in mind that the baseline was *trained* to classify 1 year olds.

In this section we report our method's performance on a real world dataset. Here the task is to detect brain Magnetic Resonance Images (MRIs) from pediatric subjects at an age (1 - 6 years) that is younger than the inlier data (9 - 11 years of age). We expect visible differences in image contrast and local brain morphometry between the brains of a toddler and an adolescent. As a

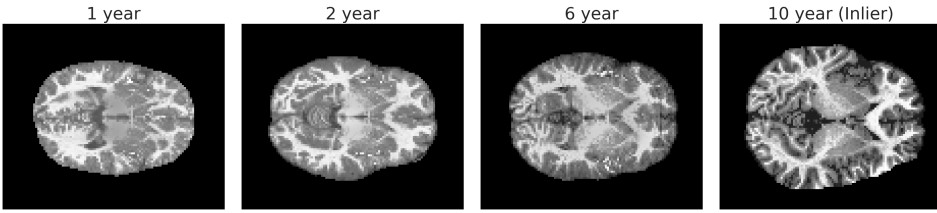

Figure 4: Note the change in image contrast, brain size and brain matter structure as the child grows. Each age is increasingly difficult to distinguish from our inliers.

child grows, their brain matures and the corresponding scans appear more like the prototypical adult brain. This provides an interesting gradation of samples being considered out-of-distribution with respect to age. We employ 3500 high resolution T1-weighted MR images obtained through the NIH large scale ABCD study (Casey et al. (2018)), which represent data from the general adolescent population (9-11 years of age). This implies that our in-distribution dataset will have high variation. After standard preprocessing, we extracted for each dataset three mid-axial slices and resized them to be 90x110 pixels, resulting in ∼11k axial images (10k training, 1k testing). For our outlier data, we employ MRI datasets of children aged 1, 2, 4 and 6 years (500 each) from the UNC EBDS database Stephens et al. (2020); Gilmore et al. (2020). Our methodology was effectively able to identify younger age groups as out-of-distribution. Table 4 reports the results for GMMs trained for this task. As expected, the separation performance decreases as age increases. Note that we kept the same hyperparameters for our auxiliary methods as in the previous experiments despite this being a higher resolution scenario. We also note that our Flow model and KD Tree perform equally as well and refer the reader to A.5. Following Liang et al. (2017) and DeVries & Taylor (2018), we compare our results to the baseline methodology proposed by Hendrycks & Gimpel (2017). We trained a standard ResNet-50 (He et al. (2016)) to classify between inliers and 1 year olds and tested its performance on unseen outliers. We observed that MSMA generally outperforms the baseline. Note that the baseline classifier is biased towards the out-of-distribution detection task since it was trained to separate 1 year olds, wheres MSMA is trained completely unsupervised. Lastly, in Section A.7 we also report results of f-AnoGAN Schlegl et al. (2019) applied on this out-of-distribution task.

## 5.5 HYPERPARAMETER ANALYSIS

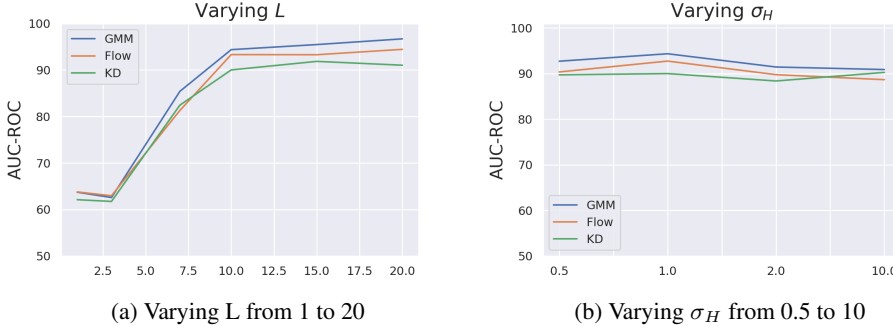

(a) Varying L from 1 to 20

(b) Varying $\sigma_H$ from 0.5 to 10

Figure 5: Analysis of the effect of hyperparameters $\sigma_H$ and $L$ on MSMA's out-of-distribution detection performance. We observe that the defaults $\sigma = 1.0$ and $L = 10$ perform the best, with a slight variance in performance when we deviate from them.

We performed experiments to analyze how sensitive MSMA is to two main hyperparameters utilized in NCSN: the number of noise levels ($L$) and the largest scale used ($\sigma_H$). We opted out of varying the smallest noise scale and kept it the same across experiments ($\sigma_L = 0.01$). Our rationale is that noise perturbations to images at such a small scale are imperceptible to the human eye and are adequate to give an estimate of the true score. For all experiments, we train on CIFAR-10 and evaluate on the *All Images* dataset described in Section 5.1 and plot AUROC in Figure 5. In order

to reduce GPU memory usage and computation time, we had to reduce the batch size from 128 to 64, which is why we see a performance dip from the main experiment in Section 5.2. We keep the same hyperparameters for our auxiliary models as in the main experiment.

For the number of scales $L$, we test the values 1, 3, 10, 15, and 20, with $\sigma_H$ fixed at the default value ($\sigma_H = 1$) . Recall that we follow the original NCSN schema by (Song & Ermon, 2019), and utilizes a geometric sequence of sigma scales from $\sigma_H$ to $\sigma_L$ with $L$ steps (endpoint inclusive). Thus, changing $L$ changes the intermediate noise scales. Our results in Figure 5a show that MSMA is optimal near the default ($L = 10$). Increasing $L$ does not significantly vary the performance, while small values such as $L = 3$ are not be adequate at providing enough intermediate scales. Note that $L = 1$ is the degenerate case where only the largest noise scale is used. This highlights the need for a range of scales as argued in Section 2.1 and empirically shows that simply using one large scale is not enough. Figure 5b plots the affect of varying the largest noise scale $\sigma_H$. We test the values 0.5, 1.0, 2.0 and 10, with the default number of scales $L = 10$. Again, we observe that our default $\sigma_H = 1$ performs the best and we do not notice any improvement form varying it. Considering how images are rescaled to [0,1] before they are passed to the network, we posit that $\sigma_H = 1.0$ already introduces large noise and increasing it further seems to degrade results to varying degrees.

Lastly, we would like to emphasize that *all* our main out-of-distribution experiments in Section5 were performed with the *same* default hyperparameters, without any tuning. Despite this disadvantage, MSMA still outperforms its competitors ODIN (Liang et al. (2017)), Confidence Thresholding (DeVries & Taylor (2018)), and Likelihood Ratios (Ren et al. (2019)), all of which need some fine-tuning of hyperparameters. Recognize that tuning requires apriori knowledge of the type of out-of-distribution samples the user expects. From the analysis in this section and our main experiment, we can confidently advocate the use of our defaults as they seem to generalize well across datasets and do not require such apriori knowledge. Admittedly, if the form of anomalies are known at training time then it would indeed be possible to tune MSMA's hyperparameters to fit a particular need. However, we leave such an analysis for future work as it starts to encroach the domain of anomaly detection whereas the work presented in this paper is mainly concerned with introducing MSMA as a performant, easy to implement, and generalizable out-of-distribution detector.

## 6    Discussion and Conclusion

We introduced a methodology based on multiscale score matching and showed that it outperformed state-of-the-art methods, with minimal hyperparameter tuning. Our methodology is easy to implement, completely unsupervised and generalizable to many OOD tasks. Even though we only reported two metrics in the main comparison, we emphasize that we outperform previous state-of-the-art in *every* metric for *all* benchmark experiments. Next, it is noteworthy that in our real-world experiment, the brain MR images are unlabeled. Since our model is trained completely unsupervised, we have to make very few inductive biases pertaining to the data.

Note that we plan to apply our method to high resolution 3D MRIs, which can reach upto 256x256x256. Consider how likelihood methods such as Glow Kingma & Dhariwal (2018) already struggle to identify out-of-distribution samples for lower resolution 2D images ( Nalisnick et al. (2018)). It is unclear whether they would preform adequately at this task when they are scaled up to the dimensionality of 3D MRIs. Furthermore, there is evidence that deep likelihood models are difficult to train in such high resolution regimes (Papamakarios (2019)), especially given low sample sizes. Our model's objective function is based on a denoising (autoencoding), and can be easily extended to any network architecture. For example, our preliminary results show that MSMA was able to achieve similar performance to Section 5.4 when applied to their full 3D MRI counterparts by utilizing a publicly available 3D U-Net (Zhou et al. (2019)). We plan to explore these high resolution scenarios as our next future direction.

Our excellent results highlight the possibility of using MSMA as a fast, general purpose anomaly detector which could be used for tasks ranging from detection of medical pathologies to data cleansing and fault detection. From an application perspective, we plan to apply this methodology to the task of detecting images of atypically maturing children from a database of typical inliers. Lastly, our observations have uncovered a peculiar phenomenon exhibited by multiscale score estimates, warranting a closer look to understand the theoretical underpinnings of the relationship between low density points and their gradient estimates.

ACKNOWLEDGMENTS

We would like to thank the UNC Early Brain Development Study (PI John Gilmore) and the Adolescent Brain Cognitive Development (ABCD) Study (https://abcdstudy.org), held in the NIMH Data Archive (NDA). The ABCD data repository grows and changes over time. The ABCD data used in this report came from DOI 10.15154/1503209 (v2.1). Other funding was provided through NIH grants HD053000, MH070890, MH111944, HD079124, EB021391, and MH115046.

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

## A APPENDIX

### A.1 DATASET DETAILS

All the datsets considered are described below.

**CIFAR-10:** The CIFAR-10 dataset (Krizhevsky et al. (2009)) consists of 60,000 32x32 colour images in 10 classes, such as horse, automobile, cat etc. There are 50,000 training images and 10,000 test images.

**SVHN:** The Street View Housing Numbers (SVHN) dataset (Netzer et al. (2011)) consists of 32x32 images depicting house numbers ranging from 0 through 9. We use the official splits: 73,257 digits for training, 26,032 digits for testing.

**TinyImageNet:** This dataset[4] is a subset of the ImageNet dataset (Deng et al. (2009)). The test set has 10,000 images with 50 samples for each of the 200 classes. Liang et al. (2017) created two 32x32 pixel versions of this dataset: *TinyImageNet (crop)* which contains random crops of the original test samples and *TinyImageNet (resize)* which contains downscaled test images.

**LSUN:** The Large Scene UNderstanding (LSUN) produced by (Yu et al. (2015)) consists of 10,000 test images belonging to one of 10 different scene classes such as bedroom, kitchen etc. Liang et al. (2017) created two 32x32 pixel versions of this dataset as well: a randomly cropped *LSUN (crop)* and a downsampled *LSUN (resize)*.

**iSUN:** This dataset was procured by (Xu et al. (2015)) and is a subsample of the SUN image database. We use 32x32 pixel downscaled versions of the original 8,925 test images.

**Uniform:** This dataset consists of 10,000 synthetically generated 32x32 RGB images produced by sampling each pixel from an i.i.d uniform distribution in the range [0,1].

**Gaussian:** These are 10,000 synthetic 32x32 RGB images where each pixel is sampled from an i.i.d Gaussian distribution centered at 0.5 with a standard deviation of 1. The pixel values are clipped to be within [0, 1] to keep the values within the expected range of (normalized) images.

**All Images**: Following DeVries & Taylor (2018), this dataset is a combination of all non-synthetic OOD datasets outlined above: *TinyImageNet (crop)*, *TinyImageNet (resize)*, *LSUN (crop)*, *LSUN (resize)* and *iSUN*. Therefore this contains 48,925 images from a variety of data distributions. Note that this collection effectively requires a single threshold for all datasets, thus arguably reflecting a real world out-of-distribution setting.

### A.2 EXPERIMENTAL DETAILS

We use the NCSN model provided by Song & Ermon (2019). In particular, we use the Tensorflow implementation provided through a NeurIPS reproducibilty challenge submission, Matosevic et al. (2019). The model architecture used is a RefineNet with 128 filters. The batch size is also fixed to 128. We train for 200k iterations using the Adam optimizer. Following Song & Ermon (2019), we use $L = 10$ standard deviations for our Gaussian noise perturbation such that $\{\sigma_i\}_{i=1}^{L}$ is a geometric sequence with $\sigma_1 = 1$ and $\sigma_{10} = 0.01$. We use the same hyperparameters for training on both CIFAR and SVHN. For our experiment on brain MRI images (Section 5.4), we trained our model with 64 filters and a batch size of 32 due to memory constraints caused by the higher resolution images. For the baseline network in Section5.4, we used the standard ResNet50 available in Keras and added a global average pooling layer followed by a dense layer of 1024 nodes followed by a softmax layer with binary output. We trained for 10 epochs using Adam with default learning rate and momentum. For unseen classes, we follow Hendrycks et al. (2018) and pick the maximum prediction probability as our outlier score. Since the network was trained to classify one year olds, we report the 1 year metrics using the logits corresponding to the outlier class only in lieu of taking the max across both classes.

We train our auxiliary models on the same training set that was used to train the NCSN model, thereby circumventing the need for a separate held out tuning set. For our Gaussian Mixture Models, we mean normalize the data and perform a grid search over the number of components (ranging

---

[4]https://tiny-imagenet.herokuapp.com/

from 2 to 20), using 10-fold cross validation. Our normalizing flow model is constructed with a MAF using two hidden layers with 128 units each, and a Standard Normal as the base distribution. It is trained for a 1000 epochs with a batch size of 128. Finally for our nearest neighbor model, we train a KD Tree to store (k=5)-nearest neighbor distances of the in-distribution training set. We keep the same hyperparameter settings for *all* experiments.

Note that in Section 5.2, as ODIN and Confidence Thresholding were trained with a number of different architectures, we report the ones that performed the best for each respective method. Specifically, we use the results of VGG13 for Confidence Thresholding and DenseNet-BC for ODIN.

## A.3 EVALUATION METRIC DETAILS

To measure thresholding performance we use the metrics established by previous baselines (Hendrycks & Gimpel (2017), Liang et al. (2017)). These include:

**FPR at 95% TPR:** This is the False Positive Rate (FPR) when the True Positive Rate (TPR) is 95%. This metric can be interpreted as the probability of misclassifying an outlier sample to be in-distribution when the TPR is as high as 95%. Let TP, FP, TN, and FN represent true positives, false positives, true negatives and false negatives respectively. FPR=FP/(FP+TN), TPR=TP/(FN+TP).

**Detection Error:** This measures the minimum possible misclassification probability over all thresholds. Practically this can be calculated as $\min_\delta 0.5(1 - \text{TPR}) + 0.5\text{FPR}$, where it is assumed that we have an equal probability of seeing both positive and negative examples in the test set.

**AUROC:** This measures area under (AU) the Receiver Operating Curve (ROC) which plots the relationship between FPR and TPR. It is commonly interpreted as the probability of a positive sample (in-distribution) having a higher score than a negative sample (out-of-distribution). It is a threshold independent, summary metric.

**AUPR:** Area Under the Precision Recall Curve (AUPR) is another threshold independent metric that considers the PR curve, which plots Precision(= TP/(TP+FP) ) versus Recall(= TP/(TP+FN) ). AUPR-In and AUPR-Out consider the in-distribution samples and out-of-distribution samples as the positive class, respectively. This helps take mismatch in sample sizes into account.

## A.4 COMPLETE RESULTS FOR EXPERIMENTS IN SECTION5.2

| In-distribution Dataset | OOD Dataset | GMM | Flow | KD Tree | ODIN | Confidence |
|---|---|---|---|---|---|---|
| SVHN | TinyImageNet | 0.0 | 0.0 | 0.0 | - | 1.8 |
| | LSUN | 0.0 | 0.0 | 0.0 | - | 0.8 |
| | iSUN | 0.0 | 0.0 | 0.0 | - | 1.0 |
| | All Images | 0.0 | 0.0 | 0.0 | - | 4.3 |
| | All Images* | - | - | - | 8.6 | 4.1 |
| CIFAR-10 | TinyImageNet | 0.0 | 0.0 | 0.3 | 7.5 | 18.4 |
| | LSUN | 0.0 | 0.0 | 0.6 | 3.8 | 16.4 |
| | iSUN | 0.0 | 0.0 | 0.4 | 6.3 | 16.3 |
| | All Images | 0.0 | 0.0 | 0.4 | - | 19.2 |
| | All Images* | - | - | - | 7.8 | 11.2 |

Table 5: Results for **FPR (95% TPR)**. *Lower* values are better.

| In-distribution Dataset | OOD Dataset | GMM | Flow | KD Tree | ODIN (DenseNet) | Confidence (VGG13) |
|---|---|---|---|---|---|---|
| SVHN | TinyImageNet | 100.0 | 100.0 | 100.0 | - | 99.6 |
| | LSUN | 100.0 | 100.0 | 100.0 | - | 99.8 |
| | iSUN | 100.0 | 100.0 | 100.0 | - | 99.8 |
| | All Images | 100.0 | 100.0 | 100.0 | - | 99.2 |
| | All Images* | - | - | - | 97.2 | 99.2 |
| CIFAR-10 | TinyImageNet | 100.0 | 100.0 | 99.9 | 98.5 | 97.0 |
| | LSUN | 100.0 | 100.0 | 99.9 | 99.2 | 97.5 |
| | iSUN | 100.0 | 100.0 | 99.9 | 98.8 | 97.5 |
| | All Images | 100.0 | 100.0 | 99.9 | - | 97.1 |
| | All Images* | - | - | - | 98.4 | 98.0 |

Table 6: Results for **AUROC**. *Higher* values are better. All three auxiliary methods perform better than baselines.

| In-distribution Dataset | OOD Dataset | GMM | Flow | KD Tree | ODIN (DenseNet) | Confidence (VGG13) |
|---|---|---|---|---|---|---|
| SVHN | TinyImageNet | 0.0 | 0.0 | 0.1 | - | 3.1 |
| | LSUN | 0.0 | 0.0 | 0.1 | - | 2.0 |
| | iSUN | 0.0 | 0.0 | 0.1 | - | 2.2 |
| | All Images | 0.0 | 0.0 | 0.1 | - | 4.6 |
| | All Images* | - | - | - | 6.8 | 4.5 |
| CIFAR-10 | TinyImageNet | 0.0 | 0.0 | 1.0 | 6.3 | 9.4 |
| | LSUN | 0.0 | 0.1 | 1.5 | 4.4 | 8.3 |
| | iSUN | 0.0 | 0.0 | 1.2 | 6.7 | 8.5 |
| | All Images | 0.0 | 0.0 | 1.2 | - | 9.1 |
| | All Images* | - | - | - | 6.0 | 6.9 |

Table 7: Results for **Detection Error**. *Lower* values are better.

| In-distribution Dataset | OOD Dataset | GMM | Flow | KD Tree | ODIN (DenseNet) | Confidence (VGG13) |
|---|---|---|---|---|---|---|
| SVHN | TinyImageNet | 100.0 | 100.0 | 100.0 | - | 99.8 |
| | LSUN | 100.0 | 100.0 | 100.0 | - | 99.9 |
| | iSUN | 100.0 | 100.0 | 100.0 | - | 99.9 |
| | All Images | 100.0 | 100.0 | 100.0 | - | 98.5 |
| | All Images* | - | - | - | 92.5 | 98.6 |
| CIFAR-10 | TinyImageNet | 100.0 | 100.0 | 99.9 | 98.6 | 97.3 |
| | LSUN | 100.0 | 100.0 | 99.8 | 99.3 | 97.8 |
| | iSUN | 100.0 | 100.0 | 99.9 | 98.9 | 98.0 |
| | All Images | 100.0 | 100.0 | 99.9 | - | 92.0 |
| | All Images* | - | - | - | 95.3 | 94.5 |

Table 8: Results for **AUPR In** with In-distribution as positive class. *Higher* values are better

| In-distribution Dataset | OOD Dataset | GMM | Flow | KD Tree | ODIN (DenseNet) | Confidence (VGG13) |
|---|---|---|---|---|---|---|
| SVHN | TinyImageNet | 100.0 | 100.0 | 99.9 | - | 99.1 |
| | LSUN | 100.0 | 100.0 | 99.9 | - | 99.6 |
| | iSUN | 100.0 | 100.0 | 99.9 | - | 99.5 |
| | All Images | 100.0 | 100.0 | 99.9 | - | 99.6 |
| | All Images* | - | - | - | 98.6 | 99.5 |
| CIFAR-10 | TinyImageNet | 100.0 | 100.0 | 99.9 | 98.5 | 96.9 |
| | LSUN | 100.0 | 100.0 | 99.9 | 99.2 | 97.2 |
| | iSUN | 100.0 | 100.0 | 99.9 | 98.8 | 96.9 |
| | All Images | 100.0 | 100.0 | 99.9 | - | 99.3 |
| | All Images* | - | - | - | 99.6 | 99.5 |

Table 9: Results for **AUPR Out** with Out-of-Distribution as positive class. *Higher* values are better

| Dataset | OOD Dataset | FPR (95% TPR) ↓ | Detection Error ↓ | AUROC ↑ | AUPR In ↑ | AUPR Out ↑ |
|---------|-------------|-----------------|-------------------|---------|-----------|------------|
| SVHN | CIFAR | 8.6 | 6.9 | 97.6 | 92.3 | 99.2 |
| | TinyImageNet (c) | 0.0 | 0.0 | 100.0 | 100.0 | 100.0 |
| | TinyImageNet (r) | 0.0 | 0.0 | 100.0 | 100.0 | 100.0 |
| | LSUN (c) | 0.0 | 0.0 | 100.0 | 100.0 | 100.0 |
| | LSUN (r) | 0.0 | 0.0 | 100.0 | 100.0 | 100.0 |
| | iSUN | 0.0 | 0.0 | 100.0 | 100.0 | 100.0 |
| | Uniform | 0.0 | 0.0 | 100.0 | 100.0 | 100.0 |
| | Gaussian | 0.0 | 0.0 | 100.0 | 100.0 | 100.0 |
| | All Images | 0.0 | 0.0 | 100.0 | 100.0 | 100.0 |
| CIFAR-10 | SVHN | 11.4 | 8.1 | 95.5 | 91.9 | 96.9 |
| | TinyImageNet (c) | 0.0 | 0.0 | 100.0 | 100.0 | 100.0 |
| | TinyImageNet (r) | 0.0 | 0.0 | 100.0 | 100.0 | 100.0 |
| | LSUN (c) | 0.0 | 0.0 | 100.0 | 100.0 | 100.0 |
| | LSUN (r) | 0.0 | 0.0 | 100.0 | 100.0 | 100.0 |
| | iSUN | 0.0 | 0.0 | 100.0 | 100.0 | 100.0 |
| | Uniform | 0.0 | 0.0 | 100.0 | 100.0 | 100.0 |
| | Gaussian | 0.0 | 0.0 | 100.0 | 100.0 | 100.0 |
| | All Images | 0.0 | 0.0 | 100.0 | 100.0 | 100.0 |

Table 10: Full results for our **GMM** experiments. ↓ indicates lower values are better and ↑ indicates higher values are better. (c) and (r) indicated cropped and resized versions respectively

| Dataset | OOD Dataset | FPR (95% TPR) ↓ | Detection Error ↓ | AUROC ↑ | AUPR In ↑ | AUPR Out ↑ |
|---------|-------------|-----------------|-------------------|---------|-----------|------------|
| SVHN | CIFAR | 10.4 | 6.9 | 97.0 | 88.3 | 99.0 |
| | TinyImageNet (c) | 0.0 | 0.1 | 100.0 | 100.0 | 100.0 |
| | TinyImageNet (r) | 0.0 | 0.1 | 100.0 | 100.0 | 100.0 |
| | LSUN (c) | 0.0 | 0.1 | 100.0 | 100.0 | 100.0 |
| | LSUN (r) | 0.0 | 0.1 | 100.0 | 100.0 | 100.0 |
| | iSUN | 0.0 | 0.1 | 100.0 | 100.0 | 100.0 |
| | Uniform | 0.0 | 0.0 | 100.0 | 100.0 | 100.0 |
| | Gaussian | 0.0 | 0.0 | 100.0 | 100.0 | 100.0 |
| | All Images | 0.0 | 0.1 | 100.0 | 100.0 | 100.0 |
| CIFAR-10 | SVHN | 8.6 | 6.8 | 96.7 | 93.4 | 97.7 |
| | TinyImageNet (c) | 0.0 | 0.0 | 100.0 | 100.0 | 100.0 |
| | TinyImageNet (r) | 0.0 | 0.0 | 100.0 | 100.0 | 100.0 |
| | LSUN (c) | 0.0 | 0.0 | 100.0 | 100.0 | 100.0 |
| | LSUN (r) | 0.0 | 0.1 | 100.0 | 100.0 | 100.0 |
| | iSUN | 0.0 | 0.0 | 100.0 | 100.0 | 100.0 |
| | Uniform | 0.0 | 0.0 | 100.0 | 100.0 | 100.0 |
| | Gaussian | 0.0 | 0.0 | 100.0 | 100.0 | 100.0 |
| | All Images | 0.0 | 0.0 | 100.0 | 100.0 | 100.0 |

Table 11: Full results for our **Flow** model experiments. ↓ indicates lower values are better and ↑ indicates higher values are better. (c) and (r) indicated cropped and resized versions respectively

| Dataset | OOD Dataset | FPR (95% TPR) ↓ | Detection Error ↓ | AUROC ↑ | AUPR In ↑ | AUPR Out ↑ |
|---|---|---|---|---|---|---|
| SVHN | CIFAR | 8.5 | 6.6 | 97.6 | 92.5 | 99.1 |
| | TinyImageNet (c) | 0.0 | 0.1 | 100.0 | 100.0 | 100.0 |
| | TinyImageNet (r) | 0.0 | 0.1 | 100.0 | 100.0 | 100.0 |
| | LSUN (c) | 0.0 | 0.1 | 100.0 | 100.0 | 100.0 |
| | LSUN (r) | 0.0 | 0.1 | 100.0 | 100.0 | 100.0 |
| | iSUN | 0.0 | 0.1 | 100.0 | 100.0 | 100.0 |
| | Uniform | 0.0 | 0.0 | 100.0 | 100.0 | 100.0 |
| | Gaussian | 0.0 | 0.0 | 100.0 | 100.0 | 100.0 |
| | All Images | 0.0 | 0.1 | 100.0 | 100.0 | 100.0 |
| CIFAR-10 | SVHN | 4.1 | 4.5 | 99.1 | 99.0 | 99.2 |
| | TinyImageNet (c) | 0.5 | 1.4 | 99.9 | 99.8 | 99.9 |
| | TinyImageNet (r) | 0.3 | 1.0 | 99.9 | 99.9 | 99.9 |
| | LSUN (c) | 0.2 | 0.9 | 99.9 | 99.9 | 100.0 |
| | LSUN (r) | 0.6 | 1.5 | 99.9 | 99.8 | 99.9 |
| | iSUN | 0.4 | 1.2 | 99.9 | 99.9 | 99.9 |
| | Uniform | 0.0 | 0.0 | 100.0 | 100.0 | 100.0 |
| | Gaussian | 0.0 | 0.0 | 100.0 | 100.0 | 100.0 |
| | All Images | 0.4 | 1.2 | 99.9 | 100.0 | 99.7 |

Table 12: Full results for our **KD Tree** model experiments. ↓ indicates lower values are better and ↑ indicates higher values are better. (c) and (r) indicated cropped and resized versions respectively

## A.5 PERFORMANCE ON BRAIN MRI

| Auxiliary Method | OOD Age (Years) | FPR (95% TPR) ↓ | Detection Error ↓ | AUROC ↑ | AUPR In ↑ | AUPR Out ↑ |
|---|---|---|---|---|---|---|
| GMM | 1 | 0.2 | 0.4 | 99.9 | 99.9 | 99.9 |
| | 2 | 0.6 | 1.0 | 99.7 | 99.5 | 99.9 |
| | 4 | 23.7 | 9.2 | 96.1 | 93.8 | 97.9 |
| | 6 | 30.5 | 9.7 | 95.0 | 92.2 | 96.8 |
| Flow | 1 | 0.2 | 0.3 | 99.9 | 99.9 | 99.9 |
| | 2 | 0.6 | 1.3 | 99.7 | 99.4 | 99.9 |
| | 4 | 12.2 | 8.4 | 97.3 | 94.6 | 98.8 |
| | 6 | 28.9 | 12.5 | 94.3 | 88.7 | 97.5 |
| KD Tree | 1 | 2.5 | 2.6 | 99.3 | 98.2 | 99.7 |
| | 2 | 3.6 | 3.1 | 98.9 | 96.2 | 99.6 |
| | 4 | 18.6 | 10.7 | 95.7 | 91.0 | 98.0 |
| | 6 | 39.2 | 14.9 | 91.6 | 84.2 | 95.8 |

Table 13: Comparison of all auxiliary models tasked to separate the brain scans of different age groups. In-distribution samples are 9-11 years of age. All values are shown in percentages. ↓ indicates lower values are better and ↑ indicates higher values are better.

## A.6 SEPARATING FASHION-MNIST FROM MNIST

Table 14 summarizes the results for our FashionMNIST (inlier) and MNIST outlier experiments. We report the same metrics as the original We observe that our results (with default hyperparameters) are not as good as their colored image counter parts. However, we are able to beat ODIN and raw likelihoods. While it may be possible to tune the hyperprameters like Likelihood Ratios, we leave that analysis for future work. Furthermore, we tried to run Likelihood Ratios using the provided code but were unable to reproduce the results presented in the original paper. Thus, we compare against their reported metrics for completeness but cannot make conclusions about their validity.

| Method | FPR (80% TPR) ↓ | AUROC ↑ | AUPR In ↑ |
|---|---|---|---|
| MSMA GMM | 27.02 | 82.56 | 77.37 |
| Flow | 31.84 | 82.05 | 80.82 |
| KD Tree | 43.27 | 69.32 | 58.43 |
| ODIN | 43.20 | 75.20 | 76.30 |
| Likelihood | 100.00 | 8.90 | 32.00 |
| Likelihood Ratios (tuned) | 0.10 | 99.40 | 99.30 |

Table 14: Comparison of auxiliary models tasked to separate CIFAR-10 (inlier) and SVHN (out-of-distribution). ↓ indicates lower values are better and ↑ indicates higher values are better.

## A.7 PERFORMANCE OF F-ANOGAN ON BRAIN MRI

We also compared our methodology to f-AnoGAN due to its promising results as an anomaly detector and its popularity in the medical community. We use the hyperparameters suggested in the original paper and trained till convergence. Our results show that f-AnoGAN does not outperform MSMA in any experiment, and is unable to match the classifier baseline reported in Section 5.4. However, we acknowledge that it is a fast method, both in terms of training and inference, and may be useful for some cases where pixel-wise anomalies are required (a current limitation of MSMA).

| OOD Age (Years) | FPR (95% TPR) ↓ | Detection Error ↓ | AUROC ↑ | AUPR In ↑ | AUPR Out ↑ |
|---|---|---|---|---|---|
| 1 | 94.0 | 24.1 | 72.4 | 80.2 | 62.2 |
| 2 | 62.7 | 13.8 | 90.6 | 92.5 | 87.0 |
| 4 | 60.9 | 19.0 | 87.9 | 88.9 | 85.8 |
| 6 | 78.1 | 31.1 | 74.6 | 76.4 | 72.4 |

Table 15: Our results show that f-AnoGAN is unable to match the performance of MSMA for this task. In fact, under some metrics such as FPR at 95% TPR, it exhibits very poor performance as an anomaly detector.

