# OpenReview forum: "Multiscale Score Matching for Out-of-Distribution Detection"
_ICLR.cc/2021/Conference — ICLR 2021 Poster_

### Official Review · AnonReviewer4 · 2020-10-21
**Interesting work on OOD detection, but could be improved by a more intuitive explanation, and more analysis.**

**Rating:** 6
**Confidence:** 4

**Review:**

Thank you for the clarifications.
I did not change my rating, since I am unclear how the proposed method compares to SOTA beyond CIFAR-10/SVHN.
Table 14 suggests that Likelihood Ratios is considerably better than the proposed method.
Furthermore, neither in Table 1, nor in Section 5.4, I can find any results of the Likelihood Ratio method.
----
The paper addresses the problem of detecting out-of-distribution (OOD) samples at test time, i.e. samples which belong to a class for which there was no training data.
For that purpose the authors propose to represent each sample x using $||s(x,\sigma_1)||, ..., ||s(x,\sigma_L)||$, where $s(x,\sigma) =  \nabla_x \log q_{\sigma}(x)$, and $q_{\sigma}$ is the the original model probability $p(x)$ + gaussian noise with variance $\sigma^2$. They call this L-dimensional space the score norm space.
The authors experimentally show that OOD samples tend to be rather distinct from in-distribution samples in the score norm space.
They exploit this, and propose to train either a Gaussian Mixture Model, Autoregressive Flow, or k-nearest neighbor model with the training data's score norm space representation.

Strong points:
- On CIFAR-10/SVHN they show that their method performs better than the Likelihood Ratios methods from (Ren et al 2019).
- On several other baseline datasets they show that their method performs better than Confidence Thresholding (DeVries & Taylor, 2018) and ODIN (Liang et al 2017).


Unclear/Weak points:

- The proposed method is quite ad-hoc. Therefore, it would be helpful to include some experimental/theoretic analysis of why the method works, and when it does not work.
The authors try to provide some intuition in Section 2.1, though the explanation seems confusing to me:
on page 2, the authors argue that a small value of $p(x)$ is not a good method to detect outlier samples (referring to Nalisnick et al 2018),
but the Toy example in Section 2.1, page 3, discusses how their method can detect samples for which $p(x)$ is low.

- The experimental results would be more convincing if their method were compared to a recent method like Likelihood Ratios (Ren et al 2019) also on other datasets than CIFAR-10/SVHN.
For example, (Ren et al 2019) also showed results for FashionMNIST/MNIST.

- How sensitive is the method to the choice of L and other hyperparameters?


Minor:
- In the reference list the authors should at least add the conference name to each publication.
- "has been observe" -> - "has been observed"
- "other unseen datasets It is important" -> "other unseen datasets. It is important"
- "loglikelihoods" -> "log-likelihoods"

---

> ### Author Response · Authors · 2020-11-24
> **Addressing the Comments of AnonReviewer4**
>
> Thanks for all the comments! We have addressed some of your concerns below.
>
> #1. We apologize for the unclear description, we updated our paper to reflect that $p(x)$ (likelihood) is indeed a good method for detecting outliers. In fact, having a very low likelihood with respect to the data distribution is a strong indicator of outlierness.
> However, Nalisnick et al. showed that current deep likelihood methods struggle to produce low likelihoods for out-of-distribution samples. We present our method as an alternative to using $p(x)$ by considering scores (gradients of $\log p(x)$ ) instead, with the added insight that we need to consider these scores at multiple scales of perturbation.
>
> #2. In Section 5.4, we compare our method against Likelihood Ratios. We have further added an additional experiment comparing our performance on FashionMNIST vs MNIST to Likelihood Ratios and ODIN in the Appendix.
>
> #3. We included a section looking at hyperparameter sensitivity. Our results show that the default hyperparameters seem to perform near-optimal, with minimal improvements after tuning. Furthermore, we would like to emphasize that all our main experiments in Section 5 utilized the same hyperparameters, regardless of the training dataset (including the brain MRI scans). This shows (at least empirically) that our defaults generalize well to different image data domains.
>
> Thank you for the helpful suggestions, we have updated the paper accordingly. Please note that we highlighted all changed sections in the revised manuscript with a colored sidebar.

---

### Official Review · AnonReviewer1 · 2020-10-27
**Official Blind Review**

**Rating:** 5
**Confidence:** 4

**Review:**

Summary:
They proposed a new  method  of OOD detection, MSMA which uses a new generative model [NCSN] and a 2nd phase of fitting a simple density model on the vector of likelihoods at the various scales.
They showed empirically good results on standard OOD image datasets (CIFAR10 vs OOD, SVHN vs OOD etc.).  They were  able to achieve perfect separation at most settings, and much improved results for CIFAR10 vs SVHN compared to previous unsupervised methods
They showed interesting application for detecting OOD in medical  images where the inliears are scans for 9-11 years of  age, and OOD  are <9 years of age.

Strength:
MSMA is straightforward, and clearly described.  Since it’s based on a fairly well tested generative model, the part of getting multi-scale likelihood from  NCSN should be fairly robust, and reproducible.
Application on medical images is novel, could potentially benefit the ICLR audience if the dataset is released.

Concerns:
#1 Robustness of method (i.e. sensitivity to hyperparameters)
MSMA introduces an auxiliary model, which introduces extra hyperparameters, e.g. number of components in GMMs.
Also, as the authors pointed out, choosing different noise scales for NCSN gives vastly different  results in terms of OOD detection.
In the multi-scale case, there is a high degree of freedom in how to choose the various noise scales.
In Figure 1b, even in the multi-scale case MNIST and FashionMNIST seemed to have overlapping score vectors. It would be good if thorough results for this pair are included in the experiment section.
A.2 presents somewhat contradicting descriptions to Section 2.1. A.2 states that all experiments are done with the largest noise scale of 10, whereas in S2.1 they said it’s only effective at a noise scale of 20.
This raises the concern of how applicable this method is to domains not studied in [NCSN].  E.g., on non-image OOD tasks e.g. those in [SEBM]. How would one choose the scale schedule in general?
Unlike Flows, VAEs,  and GANs where likelihood can be used to do model selection (e.g. using AIS), it’s unclear how to do model selection with NCSN.  This makes me wonder if the range of hyperparameters used  for  the auxiliary models is generally applicable, in the case that the base NCSN model is trained with very different hyperparameters.

#2 A somewhat restricted coverage of existing methods
Both in the introduction and conclusion the authors emphasize how MSMA is developed  with the application of medical image OOD  detection in mind.  They dismiss comparison to density methods by saying they cannot be used with their high-resolution images. This is simply not true. [Glow] can easily learn images at 256x256, whereas the images here are only 110x90.  Also, another very popular family of methods for OOD detection in medical images are those related to [fAnoGAN]. GANs are more than capable of learning images of these scales.

Providing a new and meaningful application of OOD detection such as the MRI dataset provided here is a good contribution, but it seems to me that the authors did not attempt to compare to other methods, but only tried to show MSMA somewhat works.

#3 Incomplete understanding of the method
Section 2 tries to provide some intuition about the effectiveness of the method. However, the analysis is quite brief.  Here I try to list a few questions:
 Most of the reasoning of how the  “score” is  intuitively useful is  based on how the “density” appears in the denominator.  This makes me wonder if the numerator (“gradient of the density”) is of any importance, or maybe we can improve the method if that term is removed. One obvious thing to compare to here is just train a Flow, or VAE at different noise scales and compare to them.
Figure 2 and  section 2.1 kind of explains why a large noise scale is useful, but not why using multiple scale is useful.  Why not show using a single best scale in the experiment section?
Figure 2 uses the construction of a local-model outlier to justify why large-scale is needed, but does this construction really translate to the real  world scenario?   Is it possible to show the  difference in prediction on the real datasets when using different scales, much  like in the toy setting?  If so, this method can also be  useful for selecting the local-mode outliers in the image  setting, which could inspire new applications

Minor comments:
In Section 5.4. “ … is not tackled by classifier based OOD detectors … “, this is wrong.  [Lee] and many works after does study this.
Table 2 caption is not describing Table 2

Overall, the method is simple and effective on the CIFAR10 benchmark.  It’s possible that this method is a worthy contribution. However, I’m not sure about how generally applicable this method is because I don’t see experiments in different settings, ablations studies, and/or adequate understanding of why MSMA is better than other unsupervised methods.  For the MRI task, the author  did not compare  to relevant  baselines. Lastly, the authors show no intention in open sourcing their code/dataset, which undermines the value of an empirical study.

References:
[NCSN] Song, Yang, and Stefano Ermon. "Generative modeling by estimating gradients of the data distribution." Advances in Neural Information Processing Systems. 2019.
[SEBM] Shuangfei Zhai, Yu Cheng, W. Lu, and Zhongfei Zhang. Deep structured energy based models for anomaly detection. In ICML, 2016.
[Glow] Kingma, Durk P., and Prafulla Dhariwal. "Glow: Generative flow with invertible 1x1 convolutions." Advances in neural information processing systems. 2018.
[LikelihoodRatio] J. Ren, Peter J. Liu, E. Fertig, Jasper Snoek, Ryan Poplin, Mark A. DePristo, Joshua V. Dillon, and Balaji Lakshminarayanan. Likelihood ratios for out-of-distribution detection. In NeurIPS, 2019.
[fAnoGAN] Schlegl, Thomas, et al. "f-anogan: Fast unsupervised anomaly detection with generative adversarial networks." Medical image analysis 54 (2019): 30-44.
[Lee] Lee, Kimin, et al. "A simple unified framework for detecting out-of-distribution samples and adversarial attacks." Advances in Neural Information Processing Systems. 2018.

---

> ### Author Response · Authors · 2020-11-24
> **Addressing the Concerns of AnonReviewer1**
>
> Thank you for your in-depth comments! All your feedback was appreciated. Below we have tried to answer your concerns to the best of our abilities.
>
> #1. We acknowledge the need to determine whether our scheme is sensitive to its hyperparameters. To that effect, we have included a hyperparameter analysis in the paper. Our experiments show that the defaults already perform near optimal. Additionally, we would like to emphasize that all our main experiments were performed with the same hyperparameters despite the model being trained on different datasets (consider the difference between CIFAR and brain MRIs). This shows, at least empirically, that our defaults are generalizable to different image domains. Also, we have included an analysis on FashionMNIST vs MNIST in the appendix.
>
>  In Section 2.1, we present a contrived toy example for illustrative purposes. We chose a significantly higher noise scale so that the difference in score norms was exaggerated and clearly identifiable. Further note that in this section we restricted ourselves to $L=3$ in order to plot each noise dimension. For all our main experiments, we kept the hyperparameters $L=10$ and $\sigma_H=1$. We advocate for the use of these defaults as they empirically seem to generalize well to many OOD settings. For two of our auxiliary models (GMM and Flows), we do have access to likelihoods as an easy measure to tune performance. For KD trees, one could choose the Kth neighbour cutoff point according to the largest tolerable false positive rate for the application (which would require an inlier validation set only). We hope to evaluate MSMA on non-image domains in a future work.
>
> #2. You raise a valid concern about comparisons to a baseline in our MRI experiment. We have updated the paper with results comparing MSMA to the canonical OOD detection baseline introduced by Hendrycks and Gimpel (2017). We observe that MSMA generally outperforms it and observed it to be more stable across multiple runs.
>
> While we acknowledge Glow's generative capabilities on higher resolution images, we would like to emphasize that our goal is to extend the method to high resolution 3D MRIs, those that can reach 256x256x256. We have updated the paper to reflect this intention. Under this light, it is unclear whether models such as Glow can be easily extended to those regimes with reasonable engineering and computational costs. However, our (unreported) preliminary results show that MSMA works just as well on 3D samples as it did on the 2D MRI slices reported in the paper, with the same hyperparameters. More importantly, generative models like Glow already struggle to detect out-of-distribution samples in low-resolution domains like CIFAR vs SVHN (as shown by Nalisnick et al. 2018). It is difficult to say whether the situation would improve when looking at much higher resolution 3D images.
>
> #3. We apologize for lack of clarity in Section 2. $p(x)$ is indeed important in identifying outliers but NCSN outputs gradients of $\log p(x)$ (the score). It is unclear how to remove the numerator as it is only implicitly contained in our scores. Your idea of training likelihood models at different scales is a useful comparison and we plan to pursue that research direction in the future.
>
> You raise a good question about why multiple scales are important, which may not have come across in the paper. We are not guaranteed that one scale will work for all outliers. Consider outliers close to inlier modes e.g. outliers between Low-Density outliers and Inliers in Fig 2. Our large scale results in an overlap in the score distribution of inliers and Low-Density outliers. This makes it difficult to detect the aforementioned "in-between" outliers from the inliers. However, this large scale was necessary to get a big enough neighborhood context in order to capture the further away Local-Mode outliers. Thus, all three scales in the range would be necessary. We have updated Section 2 to clarify this intuition. The idea of using multiple scales for detecting local inlier modes is indeed very interesting. We leave such an analysis for future work.
>
> #4. We are fully committed to open sourcing the code, the paper will be updated with the GitHub repo in the final version. Unfortunately, we are not allowed to redistribute the medical data. However, it is all publicly available at nda.nih.gov. As a compromise, we plan on making the model checkpoints available in the GitHub repo once the review period is over.
>
> Finally, thank you for the minor comments, we have corrected our paper accordingly. Note that we highlighted the changed sections in the revised manuscript with a colored sidebar.
>
> References
>
> Dan Hendrycks and Kevin Gimpel. A baseline for detecting misclassified and out-of-distribution
> examples in neural networks. ICLR, 2017.
>
> Nalisnick, Eric, et al. "Do Deep Generative Models Know What They Don't Know?." International Conference on Learning Representations. 2018.

---

> > ### Comment · AnonReviewer1 · 2020-11-24
> > **Thanks for the response. There are remaining concerns.**
> >
> > Thanks for the response.
> > I wish there are more results rather than just text response.
> >
> > Response #1 regarding robustness of hyperparameters is good..
> >
> > I see two problems with response #2.  A) the baseline added is the simplest possible baseline for general image data that is known to be weak.  In my original review, I suggested baselines like AnoGAN which had been shown effective on medical images.  I'm not sure how the choice of baseline was made.  B) You back up the argument that NCSN can scale to 3D data using unreported results.  Without the additional results, this claim remains unjustified.
> >
> > Other minor edits and responses are fine.  I would consider increasing the score if my remaining concerns can be further addressed.  At the moment, the section about MRI data lacks empirical support.
> >
> > Best,

---

### Official Review · AnonReviewer3 · 2020-10-28
**Very good paper on a highly relevant topic**

**Rating:** 9
**Confidence:** 4

**Review:**

#### Summary:
The authors leveraged and repurposed Noise Conditioned Score Network (NCSN) that was originally introduced by Song & Ermon (2019) for generative modeling to be used for detection out-of-distribution (OOD) images. The authors unfold the intuition and rationale behind score matching followed by the equivalence of denoising autoencoder (DAE) to derive NCSN as a score estimator and provide an analysis to demonstrate the value of multiscale score analysis. In an experimental analysis on SVHN and CIFAR datasets they demonstrate superiority of their method (MSMA) over previously reported findings in the literature using state-of-the-art models (ODIN, JEM, Likelihood Ratios) on OOD task.

##########################################################################
#### Reasons for score:
I vote for accepting. While the objective foundation of the methodology is adapted from previous work, I find the repurposing of it for fast and effective OOD detection novel and meaningful. The authors have structured and communicated their findings remarkably and provided a well designed experimental evidence to support the methodology for the detection of OOD images task.

##########################################################################
#### Pros:

1. The paper addresses a relevant issue of OOD images detection using norms of score estimates and is highly relevant to the ICLR community.

2.  The multiscale score analysis was very well done and very well communicated. The visualizations captured very well the essence of the findings and were well highlighted in in the discussion. It was clear, useful and it well justified the following method development.

3. This paper provides comprehensive experiments, well related to the scientific context, to show the effectiveness of the proposed method. The additional performance metrics in the appendix provide a well complementary supprot.

##########################################################################
#### Major comment:
While the paper is overall very well written, structured and communicated, I found the final discussion and conclusion quite lacking. 1) The claim that autoencoding taks better suits deep CNNs should be a bit more elaborated/ demonstrated. 2) The sentence on the “peculiar phenomenon exhibited by multiscale score estimates” is also not fully clear. It would be better if the authors explicitly mention to which phenomenon they relate. 3) I would find it important to add to the discussion a paragraph on the paper limitations, for example, any limitations the datasets present, limitations on the applied comparisons, limitations of the method application or others. 4) While the authors mentioned their plan to apply the methodology on a specific task, I think the discussion on future directions quite lacking. Are there other potential next steps that can be done on top of the proposed method? The analysis on range of scales mentioned in the end of section 2.1 could be an example of that. 5) As a minor suggestion, the authors may consider to relate to any wider impact of their work.

#### Minor comments:

At two points in the manuscript the authors mentioned a future application of the method to identify atypical morphometry in early brain development. Since this experimental analysis was not actually done, I found it quite distracting and out of the scope of this paper. I would therefore suggest removing it from both introduction and discussion.
Section 5.3, I would suggest to briefly mention what preprocessing was done on *_all_images_.

##########################################################################
#### Questions during rebuttal period:

Please address and above comments.

---

> ### Author Response · Authors · 2020-11-24
> **Addressing the Comments of AnonReviewer3**
>
> Thanks for all your comments! We have updated the paper to better reflect our potential next steps. One shortcoming of the method is the disjoint phases of learning i.e training the NCSN first and then the auxiliary models. We would like to explore the possibilities of joint training to improve the representations learnt by the NCSN. Additionally, we are also looking into the possibility of producing per pixel scores, which would allow us to generate heatmaps of anomalous regions in an image.
>
> Please find our updated manuscript with all changed sections and some of your suggested edits highlighted with a colored sidebar.

---

### Official Review · AnonReviewer2 · 2020-10-29
**Limited novelty, insufficient experimental or theoretical analysis**

**Rating:** 5
**Confidence:** 3

**Review:**

This paper apply multi scale score estimates to out-of-distribution detection. They demonstrate the usefulness of multi scale estimates and adopt auxiliary model to identify outlier data. The proposed method is evaluated on two different settings and is effective for out-of-distribution detection.

Strength:
+ The motivation of the proposed method is clear. The proposed method makes sense.
+ The proposed method is quite simple. Seems easy to implement.

Weakness:
1. The writing of the paper needs further improvement. This paper is based on denoting auto encoder and Noise Conditioned Score Network. But the introduction of these important works is not very clear.

2. The novelty of the method is marginal. They apply previous multi scale score estimate method on out-of-distribution detection settings. Such application is trivial.

3. Experiment settings in the paper is quite simple. The proposed method is not rigorously studied in complex datasets. The improvement of previous works for separating SVNH and MNIST is not signifiant. The method doesn't compare with previous works when applying on brain scan images.

4. Important theoretically analysis is missing. The proposed method has several important hyper parameters: number of scales, sigma value for each scale, etc. Real data distribution could be very complex, in this case, how to select these parameters? Discussions about how to the effect of these parameters are missing.

-------

Update after rebuttal:

I appreciate the efforts of providing a hyper parameter study. Thanks for the clarification about dataset used in the paper.
I would like to increase my rating from 4 to 5.  Since the proposed method is somewhat ad-hoc (shared concern among other reviewers), either experimental or theoretic analysis is important to understand when and why it works. However, I don't think these analysis are sufficient in current form.

---

> ### Author Response · Authors · 2020-11-24
> **Addressing the Concerns of AnonReviewer2**
>
> We appreciate the comments you’ve made and would like to address some of your concerns.
>
> #1. Even though these datasets are trivial to separate by humans, we want to emphasize that these experimental scenarios (e.g.  CIFAR vs SVHN) are the defacto standard in quantifying the performance for any out-of-distribution detector. Consequently, this out-of-distribution testbed has been used by [1], [2], [3], [4] and others. Moreover, Nalisnick et al. [5] showed that deep generative models such as Glow can in fact be fooled by outlying datasets, even for obvious cases such as CIFAR vs SVHN. Even methods specifically built for the purposes of out-of-distribution detection (such as ODIN) have struggled to accurately separate these easy-for-humans datasets. Therefore, we believe that achieving state-of-the-art performance in this landscape is still a worthwhile endeavor before moving on to more difficult scenarios.
>
> #2. We have added a hyperparameter analysis section (as highlighted in the manuscript). Our results show that the model is stable near our defaults, which perform near-optimal already. Furthermore, all our main experiments were run with the same defaults, showing that they do not need to be tuned on a per-dataset basis and can generalize well to different image data domains. Note the significant differences between CIFAR-10, SVHN, and brain MRI domains. Due to these reasons, we can recommend our defaults for various scenarios especially when anomalies are not known beforehand.
>
> References
>
> [1] Dan Hendrycks and Kevin Gimpel. A baseline for detecting misclassified and out-of-distribution
> examples in neural networks. ICLR, 2017.
>
> [2] Shiyu Liang, Yixuan Li, and R. Srikant. Enhancing The Reliability of Out-of-distribution Image
> Detection in Neural Networks. 6th International Conference on Learning Representations, ICLR
> 2018 - Conference Track Proceedings, jun 2017.
>
> [3] J. Ren, Peter J. Liu, E. Fertig, Jasper Snoek, Ryan Poplin, Mark A. DePristo, Joshua V. Dillon, and
> Balaji Lakshminarayanan. Likelihood ratios for out-of-distribution detection. In NeurIPS, 2019.
>
> [4] Lee, Kimin, et al. "A simple unified framework for detecting out-of-distribution samples and adversarial attacks." Advances in Neural Information Processing Systems. 2018.
>
> [5] Nalisnick, Eric, et al. "Do Deep Generative Models Know What They Don't Know?." International Conference on Learning Representations. 2018.

---

### Author Response · Authors · 2020-11-24
**Uploaded Code**

We have uploaded a zip file containing the code we used to train NCSN and our auxiliary models. Additionally, we include notebooks for our main experiments in Section 5 and the 1D GMM analysis in Section 2.

---

### Comment · ~Ahsan_Mahmood1 · 2021-03-15
**Final Version**

Thank you to all the reviewers and the ACs for their time and helpful comments! We have uploaded our final camera-ready version. For posterity, we applied f-AnoGAN to our brain MRI out-of-distribution task and report results in the appendix. Since this experiment was not performed during the review period, we leave our original analysis unchanged and simply include these results as a reference for future readers.

---

### Decision · Program_Chairs · 2021-01-07
**Final Decision**

**Decision:**

Accept (Poster)

**Comment:**

There seems to be some disagreement between Reviewers, with some borderline scores and some very good scores. After careful consideration of both reviews and answers, and after reading the updated version of the paper with some detail, I believe the approach is valuable. The use of scores for detecting out-of-distribution data is very novel and presents a number of opportunities for further research, both theoretically and empirically. Overall, my recommendation is to ACCEPT the paper. As a brief summary, I highlight below some pros and cons that arose during the review and meta-review processes.

Pros:
- Straightforward method. "Trivial application".
- Novel application to medical images.
- Robustness of default hyper-parameters.
- Future open sourcing of the code and model checkpoints.
- Topic highly relevant to the ICLR community.
- Well-written paper + relatively good visualizations.

Cons:
- Lack of comparison with other existing approaches.
- Intuition/explanation/motivation on why the method works could be improved.
- Effect of hyper-parameters could be further discussed/analyzed.
- Concerns about applicability of the approach.